# The Effect of Dimple Insole Design on the Plantar Temperature and Pressure in People with Diabetes and in Healthy Individuals

**DOI:** 10.3390/s24175579

**Published:** 2024-08-28

**Authors:** Martha Zequera, Alejandro J. Almenar-Arasanz, Javier Alfaro-Santafé, David Allan, Andrés Anaya, Mauricio Cubides, Natalia Rodríguez, Cesar Salazar, Roozbeh Naemi

**Affiliations:** 1Baspi-Footlab, Electronics Department, School of Engineering, Pontificia Universidad Javeriana, Avenida Carrera 7 41-20, Bogotá 11023, Colombia; mzequera@javeriana.edu.co (M.Z.); anayaandresj@javeriana.edu.co (A.A.); mcubides@javeriana.edu.co (M.C.); natalia-rodriguezp@javeriana.edu.co (N.R.); cesar_salazarc@javeriana.edu.co (C.S.); 2Department of Research & Innovation, Podoactiva, Technology Park Walqa, Huesca, Ctra N 330 a Km 566, 22197 Cuarte, Huesca, Spain; aalmenar@usj.es (A.J.A.-A.); javieralfaro@podoactiva.com (J.A.-S.); 3Physiotherapy Department, Campus Universitario, St. Jorge University, Autovía Mudéjar, Km. 299, 50830 Villanueva de Gállego, Zaragoza, Spain; 4School of Sport, Faculty of Life Sciences, University of Ulster, 2-24 York Street, Belfast BT15 1AP, UK; d.allan@ulster.ac.uk; 5Centre for Biomechanics and Rehabilitation Technologies, School of Health Science and Wellbeing, Staffordshire University, Leek Road, Stoke-on-Trent ST4 2DF, UK; 6Centre for Human Movement and Rehabilitation, School of Health and Society, University of Salford, Fredrick Road, Manchester M6 6PU, UK

**Keywords:** diabetic foot, thermography, orthotics, plantar pressure, foot ulcer, thermal imaging, foot temperature, dimple insole, walking

## Abstract

An increase in plantar pressure and skin temperature is commonly associated with an increased risk of diabetic foot ulcers. However, the effect of insoles in reducing plantar temperature has not been commonly studied. The aim was to assess the effect of walking in insoles with different features on plantar temperature. Twenty-six (F/M:18/8) participants—13 with diabetes and 13 healthy, aged 55.67 ± 9.58 years—participated in this study. Skin temperature at seven plantar regions was measured using a thermal camera and reported as the difference between the temperature after walking with an insole for 20 m versus the baseline temperature. The mixed analyses of variance indicated substantial main effects for the Insole Condition, for both the right [Wilks’ Lambda = 0.790, F(14, 492) = 4.393, *p* < 0.01, partial eta squared = 0.111] and left feet [Wilks’ Lambda = 0.890, F(14, 492) = 2.103, *p* < 0.011, partial eta squared = 0.056]. The 2.5 mm-tall dimple insole was shown to be significantly more effective at reducing the temperature in the hallux and third met head regions compared to the 4 mm-tall dimple insole. The insoles showed to be significantly more effective in the diabetes group versus the healthy group, with large effect size for the right [Wilks’ Lambda = 0.662, F(14, 492) = 8.037, *p* < 0.000, Partial eta-squared = 0.186] and left feet [Wilks’ Lambda = 0.739, F(14, 492) = 5.727, *p* < 0.000, Partial eta-squared = 0.140]. This can have important practical implications for designing insoles with a view to decrease foot complications in people with diabetes.

## 1. Introduction

Globally, an estimated 537 million individuals suffer from diabetes, with projections indicating that this figure will increase to 642 million by 2045 [1]. Diabetic foot ulcers (DFUs) represent a frequent complication of diabetes and can result in severe consequences, including amputations and mortality [2]. Elevated plantar pressure, caused by diminished sensitivity and foot deformity, also contributes to the risk of ulceration [3,4].

On the other hand, skin temperature referred to as a contributing factor to foot ulceration [5], and skin temperature in areas of elevated plantar pressure, have been correlated with the onset of ulcers in diabetic patients [6,7]. In a recent systematic review and meta-analyses, the use of dermal infrared thermometry to measure foot temperature, and offloading insoles were reported as the most effective interventions for preventing diabetic foot ulceration [8].

While insoles are commonly used to provide cushioning, pressure relief, and support to the feet, thereby reducing the risk of ulceration [9,10], there is a scarcity of research on the effect of insoles on reducing plantar temperature. Only one study investigated the effect of different orthopedic insoles on redistributing plantar pressure and temperature during running in healthy individuals [11].

Only the effect of active orthopedic insoles on temperature in patients with diabetic foot were investigated when a cooling unit, a mini water pump which provided cooling inside the shoe, managed to regulate the foot temperature [12]. However, the use of the pump and the attachments could be a deterring factor in usage of the system by the patients [12]. Martínez-Nova and coworkers assessed how sock design affects plantar temperature in patients with diabetic foot. The results showed an increase in plantar temperature at the heel following a prolonged exercise of 3 h [13].

While it is crucial to investigate the effects of insoles on both pressure and temperature as risk factors for diabetic foot ulceration, there is a scarcity of studies that have adequately investigated the effects of insoles on the combined temperature and pressure. In addition, only conventional insoles with smooth top cover were investigated in previous studies [11]. Insoles’ top cover can be made of dimple structures that are essentially protruded nodules designed to stimulate the foot to improve proprioception and balance [14,15]. However, the effect of these insoles on both skin temperature and plantar pressure have not been previously investigated. 

The current study aimed to assess the effect of wearing insoles with variation in the dimple top cover feature on the plantar temperature and pressure distribution. The first objective of this study was to investigate the effect of walking in dimple insoles on the plantar temperature immediately after walk. The second objective of this study was to investigate the effect of dimple insoles on the plantar pressure during walking.

## 2. Materials and Methods

A total of 26 participants F/M 18/8, aged 55.67 ± 9.58 years, including 13 individuals with type-2 diabetes (DM2) and 13 healthy individuals, were conveniently selected from the population of people with and without diabetes for participation in this study. The inclusion criterion for the diabetes group was confirmed as the presence of diabetes. Exclusion criteria were the presence of wounds and/or ulcers in the lower limbs, the presence of foot deformity or any biomechanical alterations in the lower limbs, and peripheral arterial disease or peripheral neuropathy. These were assessed through participants’ self-declaration questionnaire. The control Group consisted of healthy individuals with matched age, weight, and height characteristics as the diabetes group. Informed consent was obtained from each participant. Tests were conducted in accordance with the ethical standards established by the Declaration of Helsinki, and the study protocol was reviewed and approved by the Ethics Committee of the Faculty of Engineering at the University of Javeriana (STANDUP-14092017).

Data for the insole conditions were collected based on the protocol and details provided in Appendix A. A dimple insole (DIMPLE) had a 2.5 mm-tall dimple, while the modified dimple insole (DIMPLE-M) had a 4 mm-tall dimple. The control insole had a thin Lunasoft-shell top layer. All insoles had the same base manufactured from polyamide using 3D-HP Multi Ject printing (Appendix A). The only difference between the insole conditions was the top layer. The close-ups of the dimple and modified dimple top layers are shown in Appendix A.

The efficacy of the insole for reducing temperature was calculated as the difference between the temperature immediately after the insole usage and the baseline temperature before the insole use. 

The foot temperature was captured over the entire plantar aspect of the foot using thermal imaging protocol (Appendix A). A FLIR ONE Pro iOS for Smartphones iOS thermal imaging camera, with 19,200 (160 × 120) pixel detector resolution, was used (Appendix A). The temperature was measured over seven regions at the plantar aspect of the feet that were marked using a pen marker prior to the start of data collection (Appendix A). The seven anatomical landmarks selected as the regions of interest (ROI) included the Hallux, 1st, 3rd, and 5th metatarsal heads (as distal heads of the 1st and 5th metatarsal), the medical and lateral mid-foot, and the center of the calcaneus (Appendix A).

The foot temperature was first measured at baseline on the couch (Appendix A) after acclimatization (Table 1), following which the participant was asked to wear the shoe with the insole in (randomized order) and to walk for 20 m (Appendix A). Immediately after completing the walk, the participant was back on the couch and the foot temperature was measured at five different timeframes (0, 30, 90, 120, and 180 s) as presented in Table 2 and Table 3. This was piloted during the development of the data collection protocol, and 3 min was found to be adequate for the foot temperature to go back to the baseline temperatures (Appendix A). This was also rechecked through the study and it was ensured that the 3 min barefoot rest on the couch between insole conditions allowed the foot temperature (as appeared at 180 s in Table 2 and Table 3) to go back to the baseline figures (Table 1). Moreover, the Peak Pressure (PP) during walking was measured at the plantar aspect of the foot.

The statistical analysis was conducted using a mixed analysis of variance to identify whether there were interactions between the study variables, and whether these interactions had a significant effect on the results. Specifically, this was to investigate the effect of three different insole interventions (control, dimple, and dimple-m) on participants’ regional foot temperature (healthy or diabetic) at five different timeframes (0, 30, 90, 120, and 180 s) after walking 20 m in each insole condition. All participants used their own trainers during the experiment, and since the data from each participant were compared with the data from the same participant across three different insole conditions, the effect of the shoe would have been cancelled out.

## 3. Results

The results of the study in this section are divided by subheadings. 

### 3.1. Baseline Temerature 

The temperature at baseline is presented in Table 1. 

### 3.2. Temperature after Walking in the Insoles

The plantar temperature in the seven regions of interest is shown in Table 2 and Table 3 for the right and left feet, respectively.

### 3.3. The Effect of Insole on Plantar Temperaure

#### 3.3.1. The Effect of Insoles on Plantar Temperature in the Right Foot

The statistical analyses indicated that there was a substantial main effect for the insole condition, Wilks’ Lambda = 0.790, F(14, 492) = 4.393, *p* < 0.01, partial eta squared = 0.111. Significant differences between the insole conditions dimple and modified dimples were observed at the hallux and the third met head, as indicated in Figure 1a,c, showing the further effectiveness of the dimple insole at reducing temperature. 

There was a substantial main effect for the insole condition and status, Wilks’ Lambda = 0.662, F(14, 492) = 8.037, *p* < 0.000, partial eta squared = 0.186. The main effect comparing the temperatures across different times was significant with large effect size, suggesting differences in effectiveness across times. The main effect comparing the temperatures across different groups of participants with diabetes and healthy was significant, with large effect size, suggesting more effectiveness in the diabetes versus the healthy group.

#### 3.3.2. The Effect of Insoles on Plantar Temperature in the Left Foot

Statistical analyses indicated that there was a substantial main effect for the insole condition, Wilks’ Lambda = 0.890, F(14, 492) = 2.103, *p* < 0.011, partial eta squared = 0.056. Significant differences between the insole conditions dimple and modified dimples were observed at the hallux and the third met head, as indicated in Figure 2a,c. There was a substantial main effect for the insole condition and status, Wilks’ Lambda = 0.739, F(14, 492) = 5.727, *p* < 0.000, partial eta squared = 0.140, (see Table 1). The main effect comparing the temperatures across different times was significant, with large effect size, suggesting differences in effectiveness across times. The main effect comparing the temperatures across different groups of participants with diabetes and healthy was significant, with large effect size, suggesting more effectiveness of insoles for participants with diabetes than for those without diabetes.

### 3.4. The Effect of Insole on Plantar Presssure

#### 3.4.1. The Effect of Insole on Plantar Pressure—Right Foot

There was a substantial main effect for the insole condition, Wilks’ Lambda = 0.028, F(42, 56) = 6.593, *p* < 0.000, partial eta squared = 0.832. Significant differences between the insole conditions dimple and modified dimple were observed versus. the control insole, as highlighted in Figure 3, indicating that the overall modified dimple insoles provide better plantar pressure distribution in general at the mid-foot area. The modified dimple insole provided significantly lower peak plantar pressures at the medial mid-foot compared to the control insole, and showed lower peak plantar pressure compared to the dimple insole condition. 

No significant effect of insole condition and status was observed. The main effect comparing the pressure across different groups was not significant, indicating no difference of the effect of insoles on pressure between healthy and diabetes groups.

#### 3.4.2. The Effect of Insole on Plantar Pressure—Left Foot

There was a substantial main effect for the insole condition, Wilks’ Lambda = 0.139, F(42, 56) = 2.239, *p* < 0.002, partial eta squared = 0.627. Significant differences between the insole conditions dimple and modified dimple were observed versus. the control insole, as highlighted in Figure 4, indicating that the overall modified dimple insoles provide better plantar pressure distribution. 

The modified dimple insole provided significantly lower peak plantar pressures at the fifth met head area compared to the dimple insole condition. The modified dimple also provided lower peak plantar pressure at the medial mid-foot compared to the control insole condition.

No significant effect of insole condition and status was observed. The main effect comparing the pressure across different groups was not significant, indicating no difference of the effect of insoles on pressure between healthy and diabetes groups.

## 4. Discussion

In this study, an evaluation of the plantar pressure distribution and temperature variation was carried out in three different insoles: control, dimple, and dimple-m, in a group of 26 participants—13 controls and 13 individuals with diabetes. The results showed significant differences in the plantar temperature in hallux and third metatarsal head in both feet in dimple insole compared to modified dimple insole conditions. These locations match the locations of high shear stress that were detected on the foot during walking [6], and which are expected to create more friction and hence generate further heat in those locations. Hence, it could be hypothesized that the modified insoles would have been most effective at reducing temperature in the hallux and third metatarsal head. Although no significant difference in the ability to reduce temperature after wearing the insole was observed, compared to the conventional insoles, the dimple insole showed itself able to reduce temperature better.

In addition, the results also indicated that the modified dimple insole could reduce the peak plantar pressure in mid-foot regions compared to the conventional and dimple insoles. While these significant differences only exist in a few regions, these indicate that modified dimple insoles can have the potential to provide better pressure distribution when specifically designed for different regions of the foot. While the results cannot be directly compared to any previous studies, the results for pressure and temperature can be compared against the previous studies focusing on the interventions of design variations in insoles [16] and socks [13]. Overall, with regard to temperature, the results of this study indicated that the temperature increase in different regions of the foot can be more effectively regulated using the dimple insole compared to the modified dimple insoles.

This observation is new and complemented previous analytical studies in which the effect of insole design on thermal regulation of the foot was studied, and in which the importance of presence of air at the interface between the skin and insole was established [17]. The results of that study indicated that, in addition to the conductivity, the thermal capacity of the insole of the shoe needs to be considered [17]. The results of the current study in which the dimple insole proved to be superior to the modified dimple are in line with the previous study based on the analytical model in which it was established that different foot temperatures can be achieved when the amount of material versus the trapped air is changed in the insole design [17]. While the current experimental study can be considered as a proof of that concept, the results are encouraging and can have practical implications for the insole design for individuals with diabetes [17].

Interestingly, the result of the current study shows more effectiveness of any insoles at decreasing the temperature for participants with diabetes versus healthy individuals. This is significant and indicates that the plantar temperature can potentially be changed more effectively using different insole features in people with diabetes. This may be connected to the fact that impaired micro-circulation that happens in diabetes affects the ability of skin micro-vessels to vasodilate and to perform their thermoregulatory function [18]. In addition, the results of the effect of insole use on the plantar temperature will fade after usage, which is expected behavior as the skin temperature acclimatises to the room temperature during rest. In addition, although in this study we used a passive insole with no active way of cooling the foot down, the result achieved was a reduction in temperature of around 2 °C in a few regions of the foot. This is comparable to the results of previous studies where a similar reduction in temperature was achieved over longer walks (5 min walk, on a treadmill) using more elaborate active temperature-regulation devices [12].

In relation to plantar pressure, significant differences were observed between the conditions of the dimple and dimple-m insoles compared to the control insoles, as shown for the right foot in Figure 3f, and for the left foot in Figure 4d. It was also observed that the dimple-m insole (PV) provided an overall better plantar pressure distribution than the dimple insole in the first met head and lateral mid-foot regions, however, the results were not consistent between the left and right feet.

However, there was a consistent and significantly lower peak pressure at the medial mid-foot of both left and right feet when the modified dimple compared to the control insole was used. While a previous systematic review indicated that certain insole modifications such as arch or metatarsal support can decrease the pressure in individuals with diabetic neuropathy [16], the results of this study are encouraging. These indicate that non-conventional insoles such as modified dimple, can also potentially decrease plantar pressure in certain regions of the foot during walking.

Moreover, the effect of dimple insoles on balance was established through a systematic review of the literature in the past [15], and this study indicates that the dimple-like surface modification and design of the insoles can be used to decrease plantar pressure. This can have implications for reducing the risk of mechanical trauma to the foot.

It is also interesting to see no difference in the effect of insoles on pressure between the healthy and diabetes groups, indicating that the modified dimple (dimple m) insole worked effectively across the healthy and diabetic populations. It should also be mentioned that the effect of the insole was different between temperature and pressure, and while the dimple insole seemed to be more effective at regulating temperature in some regions, the modified dimple appeared to be more effective at reducing pressure..

In this study, the effect of insole design on plantar pressure and temperature measured on the skin was investigated. Future studies should look at designs that accommodate both the superior temperature and pressure distribution, while considering the effect of the interaction between soft tissue and dimples, in the context of effective contact area between the two. In the case of temperature, especially in two of the seven anatomical regions of interest, the hallux and third metatarsal head, the main effect comparing temperatures at different times was significant with a large effect size, suggesting differences in the longevity of the efficacy. While this can be indirectly associated with the higher shear stresses measured (during barefoot walking) in these anatomical locations [6], the shear stress needs to be measured in in-shoe conditions. Providing access to commercially available in-shoe shear sensors, future studies could investigate the effect of insole design on the relationship between temperature and shear forces during shod conditions.

It needs to be mentioned that this study investigated the immediate effect of insoles during a short walk, and the results of this study are based on a relatively small number of participants and on variations of dimple insole heights. The effect of insole designs in general, including variations in insole base or shoe condition, was not investigated, and these designs might have influenced the effectiveness of the insoles. While these can be considered a limitation of the current study, in future, the long-term effect of the entire footwear sole design on plantar temperature, pressure, and balance warrants further investigation with a larger number of participants with foot complications such as diabetic neuropathy and peripheral arterial disease.

## 5. Conclusions

The insole surface modifications were shown to be able to change the plantar skin temperature reduction capacity at a few regions of the foot including the hallux and third met head following a short walk. More importantly, the results indicate that such an effect as the reduction of temperature was significantly more prominent in participants with diabetes compared to healthy participants, indicating the greater effectiveness of insoles in the diabetic group. Moreover, insole modification can reduce peak plantar pressure in a few regions, like medial mid-foot. These findings have important implications for designing insoles for people with diabetic foot disease, with a view to reducing the temperature and avoiding foot complications.

## Figures and Tables

**Figure 1 sensors-24-05579-f001:**
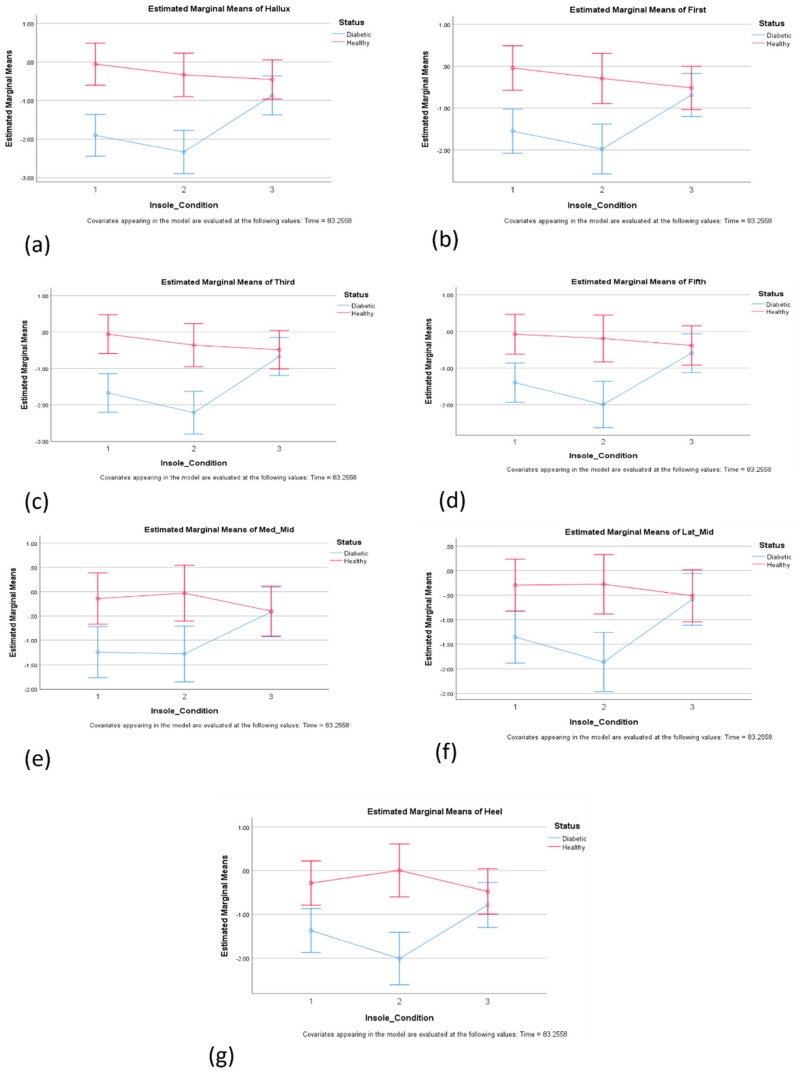
Estimated Marginal Means for temperature reduction: (**a**) hallux; (**b**) first MTH; (**c**) third MTH; (**d**) fifth MTH; (**e**) Med_Mid; (**f**) Lat_Mid; and (**g**) heel across three insole conditions 1-control, 2-dimple, and 3-modified Dimple. Significant differences were observed between the conditions of the dimple and dimple-m templates in the hallux and 3rd metatarsal head (**a**,**c**). Negative values show a reduction in temperature after use compared to the baseline.

**Figure 2 sensors-24-05579-f002:**
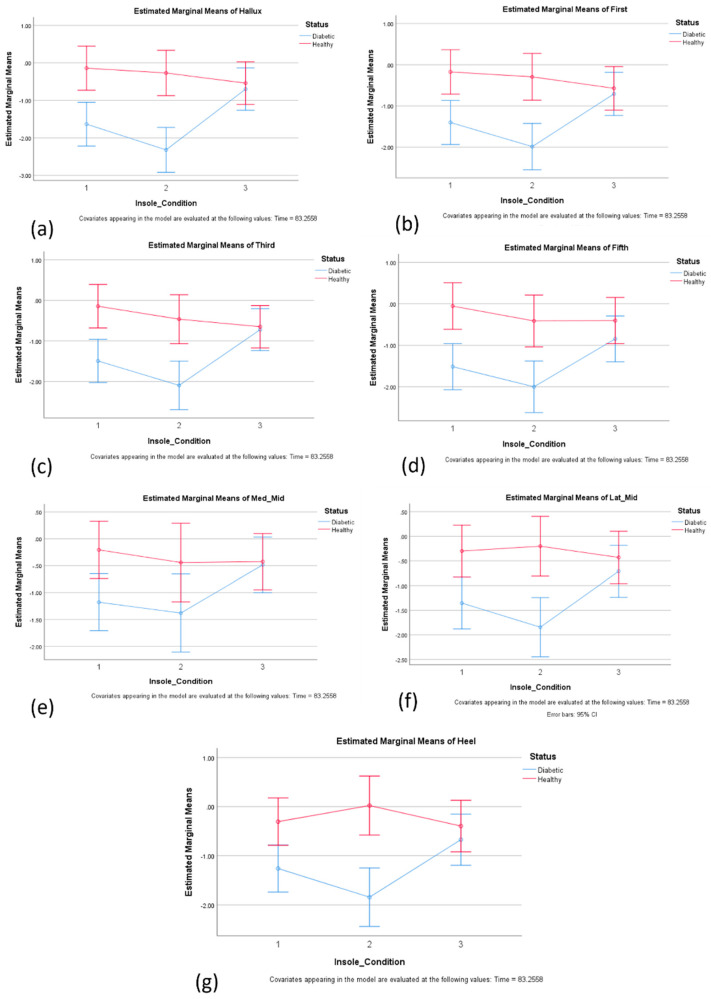
Estimated marginal means: (**a**) hallux; (**b**) first MTH; (**c**) third MTH; (**d**) fifth MTH; (**e**) Med_Mid; (**f**) Lat_Mid; (**g**) heel across three insole conditions 1-control, 2-dimple, and 3-modified dimple. Significant differences were observed between the dimple and dimple-m template conditions in the hallux and in the head of the third met, as in (**a**,**c**). Negative values show a reduction in temperature after use compared to the baseline.

**Figure 3 sensors-24-05579-f003:**
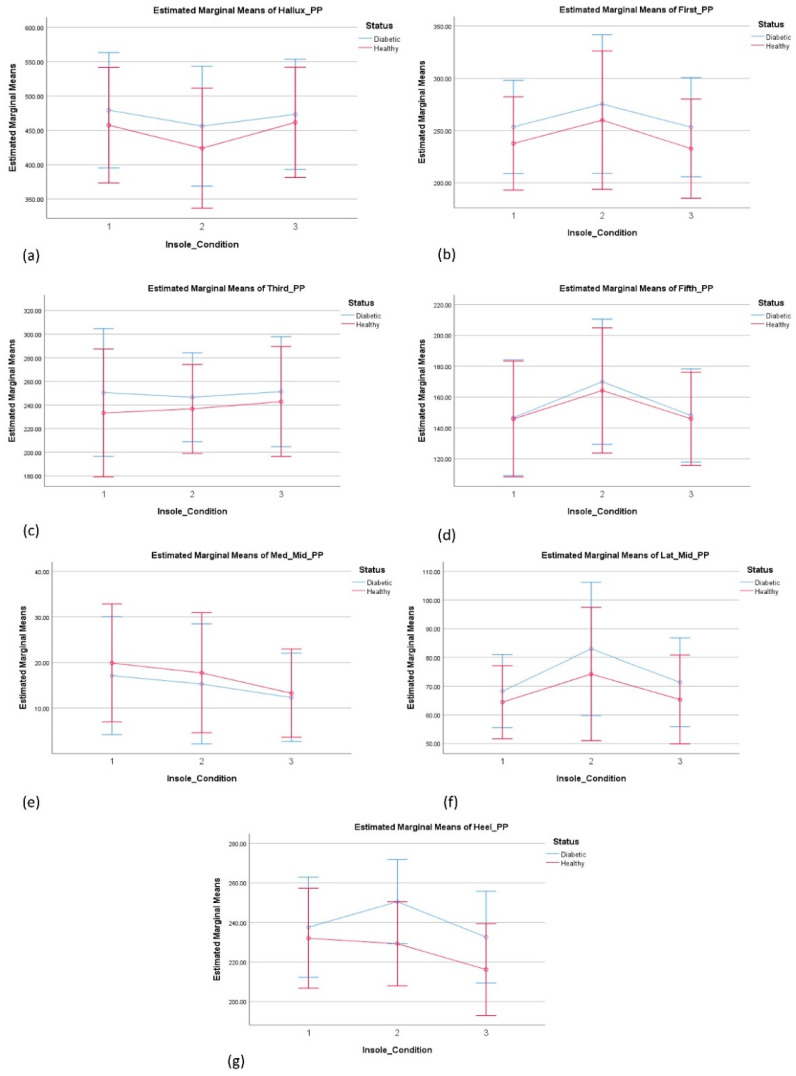
Estimated marginal means: (**a**) hallux; (**b**) first; (**c**) third; (**d**) fifth; (**e**) Med_Mid; (**f**) Lat_Mid; and (**g**) heel. The three insole conditions were 1-control, 2-dimple, and 3-modified dimple. Significant differences were observed between the conditions dimple and dimple-m at the lateral mid-foot (**f**), and between the modified dimple and the control at the medial mid-foot (**e**).

**Figure 4 sensors-24-05579-f004:**
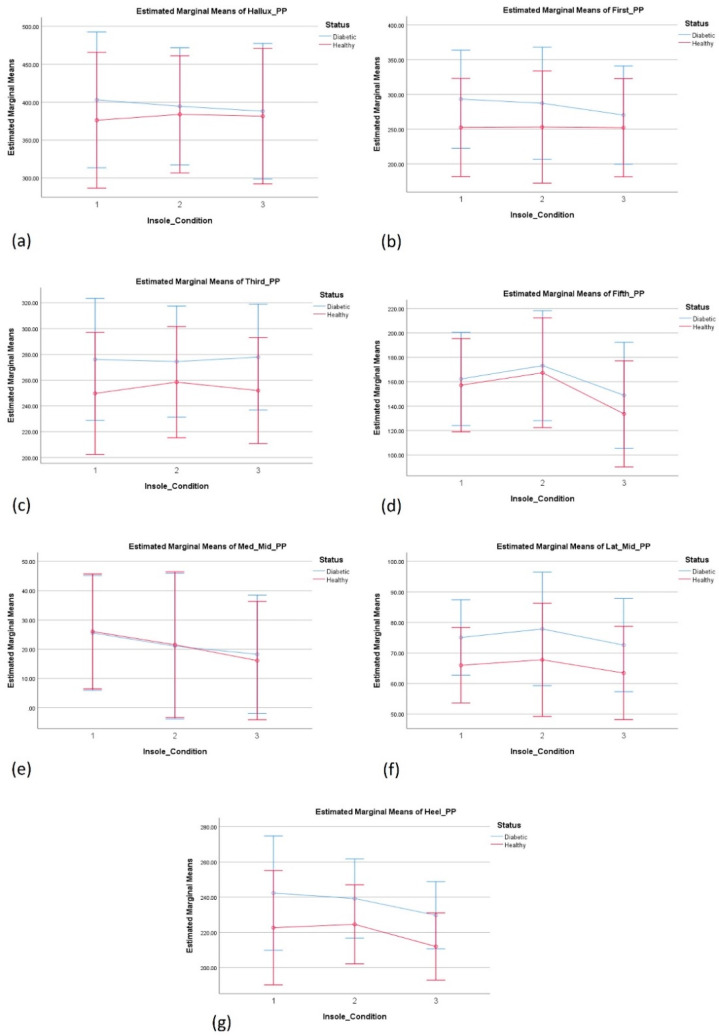
Estimated marginal means: (**a**) hallux; (**b**) first; (**c**) third; (**d**) fifth; (**e**) Med_Mid; (**f**) Lat_Mid; and (**g**) heel. The three insole conditions were 1-control, 2-dimple, and 3-modified dimple. Significant differences were observed between the conditions dimple and dimple-m at the fifth met (**d**) and between the modified dimple and the control at the medial mid-foot (**e**), and at the lateral mid-foot (**f**).

**Table 1 sensors-24-05579-t001:** The average temperature in the 7 regions of interest for the left and right feet during the baseline measurement.

Baseline Temp. °C		Hallux		1st MTH		3rd MTH		5th MTH		Medial Mid-Foot		Lateral Mid-Foot		Heel	
Status	Side	Ave	Stdev	Ave	Stdev	Ave	Stdev	Ave	Stdev	Ave	Stdev	Ave	Stdev	Ave	Stdev
**Healthy**	**Left**	22.3	3.6	23.6	3.0	23.5	3.2	22.9	3.3	24.9	2.6	23.2	2.6	22.9	3.1
	**Right**	22.4	3.4	23.6	3.0	23.7	3.0	23.0	3.2	25.1	2.5	23.6	2.4	22.8	2.6
Average-Stdev		22.4	0.1	23.6	0.0	23.6	0.1	22.9	0.1	25.0	0.1	23.4	0.3	22.8	0.1
**Diabetes**	**Left**	27.3	4.6	27.1	3.6	27.4	3.7	26.5	4.0	27.6	3.5	26.2	3.6	26.0	3.9
	**Right**	27.9	4.2	27.2	3.5	27.7	3.7	26.7	4.0	27.7	3.4	26.5	3.6	26.5	4.0
Average-Stdev		27.6	0.4	27.2	0.1	27.5	0.2	26.6	0.2	27.6	0.1	26.3	0.2	26.2	0.3

**Table 2 sensors-24-05579-t002:** The plantar temperature in the 7 regions of interest (hallux; first MTH; third MTH; fifth MTH; Med_Mid; Lat_Mid; and heel) at the right foot across the 5 timeframes (0, 30, 90, 120, and 180 s) after walking in three different insole conditions (control, dimple, and modified dimple).

Right	Temp °C		Hallux		1st Met		3rd Met		5th Met		Med Mid		Lat Mid		Heel	
Group	Insole	Time (s)	Ave	Stdev	Ave	Stdev	Ave	Stdev	Ave	Stdev	Ave	Stdev	Ave	Stdev	Ave	Stdev
Healthy	Control	0	20.4	2.0	21.6	1.9	21.7	1.9	21.4	1.8	23.3	1.9	21.8	1.9	21.1	2.2
		30	21.0	2.3	22.1	2.4	22.3	2.4	21.8	2.3	23.9	2.6	22.3	2.6	21.6	2.5
		90	20.9	2.2	21.9	2.1	22.1	2.2	21.5	2.2	23.6	2.3	22.0	2.3	21.6	2.4
		120	21.0	2.1	22.0	2.0	22.2	2.1	21.5	2.0	23.6	2.2	22.0	2.2	21.7	2.3
		180	22.0	2.4	23.3	2.2	23.4	2.4	22.9	2.3	25.0	2.6	23.4	2.5	22.8	2.4
	Ave-Stdev		21.1	0.6	22.2	0.6	22.4	0.6	21.8	0.6	23.9	0.6	22.3	0.7	21.8	0.6
Diabetes	Control	0	23.5	2.8	23.2	2.4	23.5	2.5	22.6	2.7	24.3	2.1	22.6	2.6	22.4	2.8
		30	25.1	2.5	24.9	2.2	25.1	2.2	24.3	2.5	26.0	2.2	24.3	2.5	24.0	2.8
		90	24.7	2.6	24.5	2.2	24.7	2.3	23.9	2.5	25.5	2.0	23.9	2.2	23.7	2.7
		120	24.7	2.4	24.5	2.1	24.6	2.1	23.8	2.3	25.4	1.8	23.8	2.0	23.8	2.4
		180	26.5	2.5	26.4	2.0	26.5	2.0	25.7	2.2	27.3	1.8	25.7	1.9	25.5	2.4
	Ave-Stdev		24.9	1.1	24.7	1.1	24.9	1.1	24.1	1.1	25.7	1.1	24.1	1.1	23.9	1.1
Healthy	Dimple	0	21.4	2.6	22.5	2.7	22.6	2.7	22.2	2.4	24.4	2.3	22.7	2.1	22.1	2.3
		30	21.6	2.7	22.7	2.6	22.8	2.6	22.3	2.4	24.5	2.2	22.8	2.1	22.3	2.3
		90	21.8	2.5	22.6	2.5	22.7	2.5	22.0	2.4	24.3	2.1	22.6	2.0	22.3	2.0
		120	21.3	2.5	22.2	2.5	22.2	2.4	21.5	2.3	23.9	2.1	22.2	2.0	21.9	2.0
		180	22.8	2.6	23.8	2.5	23.9	2.6	23.3	2.4	25.6	2.2	23.8	2.1	23.5	2.0
	Ave-Stdev		21.8	0.6	22.7	0.6	22.8	0.6	22.3	0.7	24.5	0.6	22.8	0.6	22.4	0.6
Diabetes	Dimple	0	23.9	4.0	23.8	3.5	24.1	3.7	23.4	3.7	25.1	3.4	23.3	3.7	23.1	4.2
		30	24.5	3.9	24.5	3.4	24.8	3.5	23.9	3.3	25.9	3.2	24.0	3.3	23.8	3.4
		90	24.5	3.9	24.4	3.3	24.6	3.3	23.9	3.2	25.6	2.8	23.8	2.9	23.8	3.3
		120	24.6	3.8	24.4	3.1	24.6	3.2	23.9	3.2	25.6	2.8	23.9	3.0	23.9	3.3
		180	26.4	3.7	26.3	2.7	26.6	2.8	25.8	2.8	27.5	2.3	25.8	2.5	25.7	3.0
	Ave-Stdev		24.8	0.9	24.7	1.0	24.9	0.9	24.2	0.9	25.9	0.9	24.2	1.0	24.0	1.0
Healthy	Modified Dimple	0	20.1	3.7	21.2	3.9	21.4	4.0	21.0	3.9	23.1	3.5	21.4	3.4	20.9	3.3
		30	21.0	3.2	22.1	3.6	22.2	3.7	21.7	3.6	24.0	3.3	22.2	3.2	21.7	3.2
		90	20.7	3.2	21.8	3.3	21.9	3.4	21.3	3.3	23.5	2.9	21.9	2.9	21.5	2.9
		120	20.3	3.1	21.4	3.3	21.4	3.4	20.8	3.3	23.1	2.9	21.3	3.0	21.2	2.9
		180	22.2	3.4	23.4	3.6	23.5	3.7	23.0	3.6	25.2	3.2	23.6	3.3	23.1	3.2
	Ave-Stdev		20.9	0.8	22.0	0.9	22.1	0.9	21.6	0.9	23.8	0.9	22.1	0.9	21.7	0.8
Diabetes	Modified Dimple	0	22.9	3.0	22.8	2.5	23.3	2.7	22.6	3.0	24.2	2.5	22.6	2.8	22.3	3.3
		30	24.0	3.1	24.1	2.7	24.5	2.9	23.7	3.2	25.4	2.7	23.8	3.1	23.5	3.4
		90	23.7	2.7	23.6	2.1	23.9	2.1	23.1	2.6	24.8	1.9	23.2	2.3	23.1	2.8
		120	23.9	2.7	23.9	2.2	24.1	2.5	23.4	2.8	25.1	2.4	23.4	2.6	23.2	2.8
		180	25.8	2.8	25.9	2.1	26.2	2.1	25.4	2.5	27.1	1.9	25.6	2.3	25.2	2.9
	Ave-Stdev		21.0	0.6	22.2	0.6	22.0	0.5	21.4	0.7	23.8	0.6	22.0	0.7	22.0	0.7

**Table 3 sensors-24-05579-t003:** The plantar temperature in the 7 regions of interest (hallux; first MTH; third MTH; fifth MTH; Med_Mid; Lat_Mid; and heel) at the left foot across the 5 timeframes (0, 30, 90, 120, and 180 s) after walking in three different insole conditions (control, dimple, and modified dimple).

Left	Temp °C		Hallux		1st Met		3rd Met		5th Met		Med Mid		Lat Mid		Heel	
Group	Insole	Time (s)	Ave	Stdev	Ave	Stdev	Ave	Stdev	Ave	Stdev	Ave	Stdev	Ave	Stdev	Ave	Stdev
Healthy	Control	0	20.3	2.3	21.6	2.0	21.4	2.2	20.9	2.1	23.2	1.8	21.4	2.0	21.3	2.5
		30	20.9	2.5	22.1	2.5	22.0	2.6	21.3	2.5	23.8	2.4	22.0	2.5	21.9	2.5
		90	20.8	2.4	21.9	2.4	21.8	2.4	21.1	2.4	23.5	2.2	21.7	2.2	21.7	2.4
		120	20.9	2.3	21.9	2.2	21.8	2.3	21.0	2.3	23.5	2.1	21.7	2.0	21.8	2.3
		180	22.0	2.6	23.2	2.5	22.9	3.0	22.6	2.4	24.9	2.4	23.1	2.4	23.1	2.4
	Ave-Stdev		21.0	0.6	22.2	0.6	22.0	0.5	21.4	0.7	23.8	0.6	22.0	0.7	22.0	0.7
Diabetes	Control	0	23.1	2.9	23.1	2.3	23.2	2.4	22.2	2.5	24.0	2.4	22.2	2.5	22.0	2.9
		30	24.7	2.4	24.7	2.1	24.8	2.3	23.8	2.7	25.7	2.4	23.7	2.7	23.6	2.9
		90	24.3	2.7	24.3	2.1	24.4	2.2	23.3	2.4	25.4	2.2	23.4	2.4	23.3	2.7
		120	24.2	2.4	24.2	1.9	24.2	2.0	23.3	2.1	25.2	2.0	23.2	2.2	23.3	2.5
		180	26.0	2.4	26.1	1.9	26.2	2.0	25.2	2.2	27.1	1.8	25.2	1.7	25.0	1.9
	Ave-Stdev		24.4	1.0	24.5	1.1	24.6	1.1	23.6	1.1	25.5	1.1	23.5	1.1	23.4	1.1
Healthy	Dimple	0	21.4	2.7	22.6	2.5	22.3	2.5	21.9	2.3	24.3	2.2	22.4	2.0	22.2	2.3
		30	21.6	2.6	22.7	2.5	22.5	2.5	21.9	2.4	24.3	2.2	22.5	2.1	22.4	2.3
		90	21.6	2.5	22.6	2.4	22.4	2.3	21.7	2.3	24.2	2.0	22.3	1.7	22.4	2.0
		120	21.2	2.5	22.2	2.3	21.9	2.4	21.2	2.3	23.7	2.0	21.8	1.8	22.0	2.0
		180	22.8	2.6	23.8	2.5	23.6	2.6	23.1	2.4	23.9	5.5	23.6	2.0	23.6	2.0
	Ave-Stdev		21.7	0.6	22.8	0.6	22.5	0.7	21.9	0.7	24.1	0.3	22.5	0.7	22.5	0.6
Diabetes	Dimple	0	23.4	4.6	23.7	3.8	23.8	4.0	22.9	4.0	24.9	3.7	22.9	3.8	22.7	4.3
		30	24.2	4.4	24.4	3.6	24.5	3.8	23.7	3.8	25.6	3.4	23.6	3.5	23.3	3.6
		90	24.0	4.2	24.2	3.4	24.3	3.7	23.3	3.7	25.4	3.2	23.4	3.4	23.3	3.6
		120	24.1	4.1	24.3	3.3	24.3	3.3	23.4	3.5	25.3	3.0	23.4	3.1	23.4	3.4
		180	26.0	3.9	26.2	2.9	26.2	3.1	25.3	3.2	27.3	2.5	25.4	2.7	25.2	3.1
	Ave-Stdev		24.3	1.0	24.5	0.9	24.6	0.9	23.7	0.9	25.7	0.9	23.7	1.0	23.6	1.0
Healthy	Modified Dimple	0	20.1	3.9	21.2	3.8	21.1	3.7	20.7	3.7	22.9	3.3	21.3	3.2	21.2	3.2
		30	20.9	3.5	22.1	3.5	21.8	3.4	21.4	3.3	23.8	3.0	22.0	3.0	21.9	3.1
		90	20.7	3.4	21.7	3.3	21.6	3.3	21.0	3.4	23.3	2.8	21.6	2.7	21.7	3.0
		120	20.4	3.4	21.3	3.3	21.1	3.3	20.4	3.4	22.9	2.8	21.1	2.8	21.4	3.1
		180	22.3	3.7	23.4	3.5	23.1	3.5	22.7	3.5	25.0	3.2	23.3	3.1	23.3	3.3
	Ave-Stdev		20.9	0.8	21.9	0.9	21.7	0.8	21.2	0.9	23.6	0.9	21.9	0.9	21.9	0.8
Diabetes	Modified Dimple	0	22.7	3.0	22.8	2.4	23.0	2.6	22.3	2.6	23.8	2.5	22.2	2.5	21.9	3.0
		30	23.9	2.9	24.1	2.4	24.3	2.5	23.4	2.5	25.2	2.3	23.4	2.3	23.1	2.6
		90	23.5	2.7	23.6	1.9	23.7	2.1	22.8	2.1	24.6	1.9	22.8	1.8	22.7	2.2
		120	23.7	3.0	23.8	2.3	23.9	2.4	23.0	2.5	24.8	2.2	23.0	2.0	22.8	2.2
		180	25.6	2.8	25.8	2.0	26.0	2.2	25.2	2.2	26.9	1.8	25.1	1.8	24.9	2.1
	Ave-Stdev		23.9	1.1	24.1	1.1	24.2	1.1	23.3	1.1	25.1	1.1	23.3	1.1	23.1	1.1

## Data Availability

Data are available upon reasonable request.

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
