# Peer review of "The Effect of Dimple Insole Design on the Plantar Temperature and Pressure in People with Diabetes and in Healthy Individuals"

_sensors, 2024, doi:10.3390/s24175579_

Round 1

Reviewer 1 Report

Comments and Suggestions for Authors

The topic is interesting, but some key issues can be clarified in detail to provide valuable information to advance the knowledge in plantar temperature and pressure for insole evaluations:   

  1. As shown in abstract and introduction, the key focus of this study is to compare the influence on dimple insole designs with protruded nodules of 2.5mm (DIMPLE) and 4mm (modified DIMPLE-M). It is suggested to modify the manuscript title in order to align with the focus of this study
  2. Following point #1 above, the structural design and material of the insoles are not found in the manuscript.  As plantar temperature and pressure will be greatly affected by insole structures and material, please provide clear specifications of the 3 insole conditions, their thickness, compressive properties, shape, size, density of the dimples, etc. 
  3. Thermal camera was the key equipment adopted in this study.  If one single point was used as the representative measurements of the plantar region?  How the points can be consistently and precisely located amongst multiple measurements? If repeated measurements have been taken?
  4. The walkway is relatively short while temperatures at time intervals 0 to 180 sec were taken.  It is noted that foot skin temperature changes are relatively slow.  Subjects were generally invited to walk for 15-20 minutes until the foot temperature was stable, in past studies.  Why authors adopted only 3 minutes as foot skin temperature may increase continuously.
  5. It is also noted that the foot skin temperature dropped very slowly when one insole experiment has been completed.  Hence, for experiments with multiple insole conditions, subjects may only complete one insole test at each attempt to ensure the foot condition is suitable.  Please elaborate how data is suitably controlled in this study for comparison. 
  6. Please discuss the possible parameters that affecting the temperature and plantar results obtained from the experiment.
  7. Authors are suggested to discuss the limitations of this study. 

Comments on the Quality of English Language

minor editing may be required

Author Response

Thanks for your positive feedback and constructive comments. Please see below the response to each point and indication of where the manuscript is revised as attached. 

Reviewer 2 Report

Comments and Suggestions for Authors

Please review and comment on each of these questions within the text of your article, complement them or list them as limitations of the study.

1. This study has limitations because it was a study conducted with a small number of participants, which may not represent the general population. In addition, the participants knew they were being studied, which could have affected their behaviors or results, known as the "Hawthorne effect".

2. The researchers did not adjust for some confounders. For example, they did not consider the participants' daily physical activity, which can influence plantar temperature and pressure. Another unadjusted confounder was diet, which can also affect foot health, especially in people with diabetes.

3. The authors also failed to adjust the results for these confounders. They did not use statistical methods to control for these variables that could affect the results, such as multiple regression. This means that the observed differences in plantar temperatures and pressures may be partially caused by other factors unrelated to the type of insole. This makes it more difficult to determine whether the observed changes were actually caused by the insoles or by other factors.

4. The study did not specifically control for HbA1c levels, which are important for assessing long-term blood glucose control in diabetic patients. How did the study account for the variability in HbA1c levels among diabetic participants, which can affect peripheral neuropathy and foot temperature? High HbA1c levels are associated with greater risks of neuropathy and complications, which could influence the study's outcomes on foot temperature and pressure. The authors should consider discussing this limitation and its potential impact on the results.

5. Did the study measure the long-term effects of using different insoles on muscle strength and balance in the lower limbs?

6. The study focused on immediate effects after a short walk but did not assess long-term outcomes like muscle strength and balance. These factors are crucial for understanding the full benefits or drawbacks of using specific types of insoles over time. The authors could mention this limitation and suggest it as an area for future research.

7. Was there an evaluation of the structural deformities of the participants' feet (e.g., flat feet, high arches) before the intervention? The study did not include an assessment of foot structural deformities, which can significantly impact pressure distribution and the effectiveness of insoles. Different foot structures may require different insole designs for optimal support. This limitation should be acknowledged, and the authors could suggest including such evaluations in future studies.

8. Were the participants' footwear standardized during the tests, and if not, how might this have influenced the results? The study allowed participants to use their own trainers, which could introduce variability in the results due to differences in footwear support and fit. This lack of standardization is a potential confounder that could affect the study's findings. The authors should address this issue and discuss its implications in the discussion section.

9. The study did not detail the mechanical properties or quality control measures for the insoles used. How was the material and mechanical properties of the insoles evaluated to ensure consistent quality and performance? Variations in material properties could affect the outcomes related to pressure distribution and temperature. The authors should discuss the importance of material consistency and suggest including these details in future studies.

10. The study did not account for dietary habits, which are important for managing diabetes and overall foot health. Did the study consider the participants' dietary habits, which can influence foot health and the risk of complications? Diet can affect inflammation, circulation, and healing, all of which are relevant to the study's focus on foot temperature and pressure. The authors could include a note on this limitation and propose investigating dietary impacts in future research.

11. Was there an analysis of the cost-effectiveness of using these specialized insoles for preventing diabetic foot complications?

12. The study did not evaluate the cost-effectiveness of the insoles, which is crucial for public health applications. Understanding the economic feasibility of widespread insole use is important for healthcare policy and recommendations. The authors should consider discussing the potential economic implications and the need for future cost-benefit analyses.

13. The study did not mention calibration or validation of the thermal imaging camera, which could affect the accuracy of temperature measurements. Were there any calibration or validation procedures used for the thermal imaging camera to ensure accurate temperature readings? Proper calibration is essential to avoid measurement errors. The authors should address this oversight and emphasize the importance of calibration in future studies.

14. Complete the text with the technical specifications of the thermal imaging camera, such as detector resolution, temperature range, accuracy (°C), thermal sensitivity (mK), field of view - FOV, spatial resolution (mrad), frame rate (Hz), certifications, NEDT, and lens grade. As well as the model of the cell phone used since it is not a medical device but a sensor connected to a cell phone.

15. Did the study control for potential biases like selection bias or measurement bias in its design and analysis? The study did not clearly outline methods to control for potential biases such as selection bias (e.g., non-random participant selection) and measurement bias (e.g., inconsistency in temperature measurement). These biases could influence the validity of the study's findings. The authors should discuss these potential biases and consider how they might have impacted the results. They could also suggest ways to minimize these biases in future studies.

16. Although it is possible to find in the literature in many old papers, the terms "hyperthermia" and "hypothermia" are nowadays incorrect in this context because the temperatures recorded by the authors do not reach levels that are normally perceived as hot or cold by people. Typically, something is considered hot when temperatures exceed around 40°C and cold when below 15°C. The highest temperature observed in the study was 27.9°C, which would not generally be felt as hot by most people and be confusing to the readers. Nobody will feel hyperthermic if touch something at 27.9°C. Furthermore, the terms "hyperthermia" and "hypothermia" are used to describe abnormal core body temperature conditions, not peripheral temperatures like those measured on the feet. Central body temperature reflects the core temperature of the body, which is critical for diagnosing conditions like fever (hyperthermia) or hypothermia due to cold exposure. In this study, the correct terminology should be "areas of higher or lower temperature" or "warmer or cooler areas," rather than "hyperthermic" or "hypothermic." These terms accurately describe the variations in temperature without implying a clinical condition related to central body temperature.

Author Response

Thanks very much for your comments and constructive feedback. Please see the response and indication of where the content is added to the manuscript in the attached. 

Round 2

Reviewer 1 Report

Comments and Suggestions for Authors

Thank you for the revised manuscripts.  It is suggested to emphasize that the "temperatures" obtained are just foot skin temperatures.  On the other hand, the manuscript mainly focuses on the foot skin temperatures and plantar peak pressures.  As the dimple design may affect the contact area between foot plantar and insole, if contact area should also be reviewed?  In Discussion section, authors are suggested to explain the results.  For example, what are the possible reasons for the significant differences in the plantar temperature in hallux and the 3rd MTH between the dimple and dimple-M?  Why dimple-M can reduce the PP at mid-foot regions?  And, why the results were not consistent between the left and the right feet?  If the differences are statistically insignificant?

It is also interesting that the results of the effect of insole use on the plantar temperature will fade after usage as the skin temperature acclimatise to the room temperature during the rest.  Hence, what is the room temperature?  If the room temperature and humidity were strictly controlled throughout the test?

Author Response

Reviewer 1

Thank you for the revised manuscripts.  It is suggested to emphasize that the "temperatures" obtained are just foot skin temperatures.  On the other hand, the manuscript mainly focuses on the foot skin temperatures and plantar peak pressures.  As the dimple design may affect the contact area between foot plantar and insole, if contact area should also be reviewed? 

Thanks for this comment. This is now added to the manuscript as a point for future research.

“In this study the effect of insole design on plantar pressure and temperature measured at the skin was investigated. Future studies should look at designs that accommodate both the superior temperature and pressure distribution while considering the effect of the in-teraction between soft tissue and dimples and the effective contact area between the two. In the case of temperature, especially in two of the seven anatomical regions of interest, hallux and third metatarsal head, the main effect comparing temperatures at different times, was significant with a large effect size, suggesting differences in the longevity of the efficacy. While this can be indirectly associated to the higher shear stresses measured during barefoot walking in these anatomical locations[6], the shear stress need to be measured during in-shoe conditions. Providing access to commercially available in-shoe shear measurement sensors, future studies can investigate the effect of insole design on the relationship between temperature and shear forces during shod conditions.”

In Discussion section, authors are suggested to explain the results.  For example, what are the possible reasons for the significant differences in the plantar temperature in hallux and the 3rd MTH between the dimple and dimple-M?  Why dimple-M can reduce the PP at mid-foot regions?  And, why the results were not consistent between the left and the right feet?  If the differences are statistically insignificant?

                These are now added to the manuscript in the Discussion section and referrals were made to previous studies where the highest shear forces and hence friction and heat generation were observed in these regions. This is now added as follows:

“These locations match the locations of high shear stress that were detected at the foot dur-ing walking [6] that are expected to create more friction and hence further heat generated in those locations. Hence it could be hypothesized that the modified insoles would have been most effective in reducing temperature in the Hallux and third metatarsal head.”

It is also interesting that the results of the effect of insole use on the plantar temperature will fade after usage as the skin temperature acclimatise to the room temperature during the rest.  Hence, what is the room temperature?  If the room temperature and humidity were strictly controlled throughout the test?

                All measurements were done in a temperature controlled room which was centrally controlled at 21 degrees the humidity was controlled centrally. This is now added to the manuscript in the Method section in the appendix.

Reviewer 2 Report

Comments and Suggestions for Authors

I am pleased to inform you that your manuscript is now in better condition for publication following a thorough review. The revisions have addressed many of the earlier issues, substantially raising the overall standard and clarity of the paper to the readers. But please there are more 2 points you need to review before to publish it.

If the device was not calibrated because the manufacturer's instructions stated that calibration is unnecessary, for scientific reasons, you must explicitly mention in the article that the FLIR One Pro device has an accuracy error of 5% in its measurements, as shown in the datasheet of the device explaining the accuracy specifications.

Additionally, please include the temperature scale in the thermal image of Figure 8, just as it was presented in Figure 7. This is essential for readers to clearly and objectively understand and analyze what each color represents in terms of temperature.

Author Response

Reviewer 2

I am pleased to inform you that your manuscript is now in better condition for publication following a thorough review. The revisions have addressed many of the earlier issues, substantially raising the overall standard and clarity of the paper to the readers. But please there are more 2 points you need to review before to publish it.

Thanks very much for your comments and constructive feedback. Please see the response and indication of where the content is added to the manuscript as follows. 

If the device was not calibrated because the manufacturer's instructions stated that calibration is unnecessary, for scientific reasons, you must explicitly mention in the article that the FLIR One Pro device has an accuracy error of 5% in its measurements, as shown in the datasheet of the device explaining the accuracy specifications.

            Thanks for this comment. The accuracy is now added to the methodology section of the manuscript appendix to ensure that the reader considerers this when using the device.

 “Please note that according to the manufacturer’s specifications, the accuracy is ±3°C (5.4°F) or ±5%, that is the typical percent of the difference between ambient and scene temperature. This is mentioned to be applicable 60 sec after start-up when the unit is within 15°C to 35°C (59°F to 95°F) and the scene is within 5°C to 120°C (41°F to 248°F). These all need to be considered when operating the system.”

Additionally, please include the temperature scale in the thermal image of Figure 8, just as it was presented in Figure 7. This is essential for readers to clearly and objectively understand and analyze what each color represents in terms of temperature.

                        This is now presented in Figure 8 and another image is added to show the exact locations of measurement along with the legend to show the temperature scale.